# Modulation of the Translation Efficiency of Heterologous mRNA and Target Protein Stability in a Plant System: The Case Study of Interferon-αA

**DOI:** 10.3390/plants11192450

**Published:** 2022-09-20

**Authors:** Alexander A. Tyurin, Orkhan Mustafaev, Aleksandra V. Suhorukova, Olga S. Pavlenko, Viktoriia A. Fridman, Ilya S. Demyanchuk, Irina V. Goldenkova-Pavlova

**Affiliations:** 1Timiryazev Institute of Plant Physiology, Russian Academy of Sciences, ul. Botanicheskaya 35, 127276 Moscow, Russia; 2Bioengineering Laboratory of the Excellence Center for Research, Development and İnnovations, Baku State University, Zahid Khalilov str., 33, Baku AZ1148, Azerbaijan

**Keywords:** interferon-αA, regulatory sequences, thermostable lichenase, protein-stabilizing partner, plant farming

## Abstract

A broad and amazingly intricate network of mechanisms underlying the decoding of a plant genome into the proteome forces the researcher to design new strategies to enhance both the accumulation of recombinant proteins and their purification from plants and to improve the available relevant strategies. In this paper, we propose new approaches to optimize a codon composition of target genes (case study of interferon-αA) and to search for regulatory sequences (case study of 5′UTR), and we demonstrated their effectiveness in increasing the synthesis of recombinant proteins in plant systems. In addition, we convincingly show that the approach utilizing stabilization of the protein product according to the N-end rule or a new protein-stabilizing partner (thermostable lichenase) is sufficiently effective and results in a significant increase in the protein yield manufactured in a plant system. Moreover, it is validly demonstrated that thermostable lichenase as a protein-stabilizing partner not only has no negative effect on the target protein activity (interferon-αA) integrated in its sequence, but rather enhances the accumulation of the target protein product in plant cells. In addition, the retention of lichenase enzyme activity and interferon biological activity after the incubation of plant protein lysates at 65 °C and precipitation of nontarget proteins with ethanol is applicable to a rapid and inexpensive purification of fusion proteins, thereby confirming the utility of thermostable lichenase as a protein-stabilizing partner for plant systems.

## 1. Introduction

Production of recombinant proteins is one of the key goals in biotechnology. The most relevant problem in this area is to produce polypeptides, the natural source of which is very limited. As a rule, these are various human and animal proteins of pharmaceutical importance. For this purpose, valuable polypeptides are produced via their biosynthesis in different heterologous systems, either prokaryotic or eukaryotic. Hundreds of proteins are successfully produced in plants, including pharmaceutical proteins, such as antibodies, vaccines, hormones, and enzymes, as well as proteins for diagnostic, research, and cosmetic purposes. A unique specific feature in the expression of plant proteins is a diversity of plant species and systems used for their production. Various plant molecular farming (PMF) technologies, such as nuclear expression, chloroplast expression, virus transfection, and transient experiment, underlie considerable success in PMF, allowing the produced proteins to meet industrial and clinical standards [1,2,3].

As compared with common expression systems used for the production of pharmaceutical and nonpharmaceutical products, plants have both economic and technical advantages. The rapidly developing area of PMF first and foremost utilizes infiltration methods to construct transient modified plant tissues by introducing agrobacteria carrying foreign genes. Thus, agroinfiltrated tissues become biofactories for producing recombinant proteins [4]. According to the current opinion, the most important advantage of transient expression systems is a high production rate, making it possible to manufacture recombinant proteins just during several days. This is of paramount importance in the case of urgent vaccines and diagnostic tools that are necessary to obtain in several weeks or months after the confirmation of a virus gene sequence [2].

Despite current achievements in the PMF area, there are still certain limitations in the development of pharmaceuticals of plant origin; first and foremost, this refers to a low yield of plant systems and a high cost of the optimization of production and purification stages [5]. According to current opinion, low yields of the recombinant proteins produced in plant systems still present the most serious problems. These problems are first and foremost determined by insufficient or excessive efficiency of the key cellular activities involved in heterologous gene expression, such as transcription, translation, posttranslational modification, and proteolytic degradation [3,6]. Great efforts have been made to improve expression systems and develop alternative strategies allowing for an increase in both the amount and quality of recombinant proteins produced in plant systems.

The expression efficiency of heterologous genes (from a gene nucleotide sequence to an active protein) is, as a rule, regulated at the stages of transcription, translation, and stability of the corresponding protein product. Currently, researchers can strictly control the expression level of heterologous genes mainly at the stage of transcription by utilizing well-studied promoters. Even if the transcription activity of a heterologous gene is maximized, this does not guarantee a high accumulation level of the protein product in the plant. A high level of mRNA of a heterologous gene must be supplemented by an efficient protein synthesis, and this has emerged to be a much more complex problem than was expected [7,8]. The question is how the mRNA translation efficiency of a heterologous gene can be increased. Translation is an intricate biological process involving a large number of players. In this process, the mRNAs themselves carry the regions that modulate translation at certain important “checkpoints” of translation initiation and elongation, such as the 5′-untranslated region (5′UTR) and coding region (CDS) [8]. According to current opinion, 5′UTR has numerous regulatory elements able to determine the further fate of any transcript in the process of translation, and this assertion has its experimental confirmations [7,9]. In particular, attachment of the 5′UTR of Arabidopsis alcohol dehydrogenase to recombinant human β-glucocerebrosidase provided a better yield of recombinant protein as compared with the tested *Nicotiana benthamiana* expression constructs [3,10]. Another important mechanism involved in the control of translation at the stage of elongation is the phenomenon of codon bias, which may result in a poor expression of a gene from one (initial) organism in another (host) organism because of a prevalence of unfavorable codons [11]. Currently, it is experimentally confirmed that the yield of recombinant protein can be increased via the maximization of codon preferences (replacement of each codon in mRNA with the preferred host codon) or harmonization of the codon preference (replacement of each codon in mRNA with an equivalent codon in terms of its usage rate in the host organism) [11].

Another important problem for the scientific community dealing with PMF technology is an increase in the yield of recombinant proteins by minimizing the degradation of heterologous proteins. For this purpose, it is necessary to design efficient strategies for preventing proteolytic degradation of recombinant proteins in plants, which primarily demands knowledge about the materials underlying this process in plant objects [12]. Note that any living organism has a well-adjusted process for protein degradation into fragments that are further used by ribosomes to assemble the necessary proteins. The lifespan of proteins in an organism is determined by their particular role; however, the lifespan of over 20% of the proteins in an organism is from several hours to several days. The proteins carry specific degradation signals, which are complex and diverse since these signals mark not only the proteins to be eliminated by proteolysis, but also the time of their elimination and the rate of their proteolytic cleavage. Eukaryotic cells have the ubiquitin conjugation system to recognize and decode signals of these types [13]. The corresponding mechanism underlying the operation of this system follows the N-end rule. This rule asserts that the protein half-life is determined by a specific N-terminal amino acid residue of its polypeptide chain [13]. So far, the N-end rule has been studied to a considerable degree with the help of a set of artificial substrates. This approach has provided deep insight into many general principles, including the lists of stabilizing and destabilizing amino acid residues at the second position in a protein sequence, which in many respects determine protein stability. Nonetheless, the N-end rule has almost never been applied, in particular, aiming to increase the number of recombinant proteins produced in plant systems.

In order to increase the accumulation of recombinant proteins in plants, the strategy of protein fusion with a stabilizing partner has been frequently used. This strategy implies construction of the hybrid genes with transcriptional–translational fusion of a target gene and the sequence of a gene coding for a protein-stabilizing partner (PSP) [2,3,12,14,15]. Numerous studies have demonstrated that the production of recombinant proteins fused to a PSP in plants has a positive effect on their accumulation. So far, researchers have an impressive list of PSPs at their disposal [2,3,12,14]. However, none of the PSPs used for fusion with target proteins is able to universally interact with the partner protein, although the list of PSPs is impressively long. This problem forces researchers to search for new PSPs and test them [15], including variants applicable not only to terminal fusions to target genes, but also to create fusion proteins via insertion of domains. This is associated with the fact that the strategy of a direct fusion may well fail because of structural instability and an increased sensitivity of such fusion proteins to proteolytic degradation [15,16]. A domain is inserted by integrating one protein domain (guest domain) into another (host domain), namely, into the region of integration sites carefully selected in the latter [16]. The current experimental data suggest that insertion fusion of proteins is more advantageous with respect to the stabilizing effects on recombinant proteins [16]. Earlier, we proposed a thermostable lichenase, β-1,3-1,4-glucanase (lichenase), i.e., endo-β-1,3;1,4-glucan-D-glucosyl hydrolase (EC 3.2.1.73; P29716) of *Clostridium thermocellum*. The structure of the protein molecule as well as its high thermostability and specific activity make this enzyme most attractive for constructing fusion proteins [17,18]. In addition, we demonstrated earlier that the insertion of interferon-αA into a loop (53 amino acid residues, aa) of *C. thermocellum* thermostable lichenase gives an efficient expression of the soluble recombinant protein form in *E. coli* periplasm without compromising the interferon-αA biological activity or the lichenase activity and thermostability. Moreover, we showed that lichenase retains its activity during ethanol precipitation of target proteins, suggesting that lichenase can act as a solubility enhancer and be useful in the rapid and efficient purification of fusion proteins [16].

In this work, we (i) tested new in silico approaches to optimize the codon compositions in target genes (case study of interferon-αA) and to the search for regulatory sequences (case study of 5′UTR) aiming to modulate a key cellular process, translation, and, correspondingly, to attain an efficient synthesis of protein products by the target genes in plants; (ii) assessed the feasibility of stabilizing the protein product according to the N-end rule or by the use of a new PSP (*C. thermocellum* thermostable lichenase) in a plant system; and (iii) estimated the adequacy of the thermostable lichenase for the rapid and efficient purification of target proteins.

## 2. Results and Discussion

### 2.1. Constructing Efficient Systems for Expression of Target Polypeptides in Plants

#### 2.1.1. Modulation of Translation Efficiency: The Role of Codon Composition and 5′UTR of Heterologous Gene

##### In Silico Analysis

The first part of the in silico study comprised analysis of the nucleotide sequence of the target gene (in our case, interferon-αA gene) and its modification with the help of JetGene software (see Materials and Methods). Initially, we assessed the distribution of codon frequencies of all *A. thaliana* genes. Then, the codon distribution in the target gene was compared to that in the overall pool of *A. thaliana* genes. Finally, the relatively “rare” triplets for individual amino acids in the target gene sequence were replaced with the triplets represented with higher frequencies in the overall pool of plant genes to equalize the codon frequencies in the target interferon-αA gene and the overall pool of all model plant genes (Appendix A). Thus, the in silico analysis allowed us to deduce the nucleotide sequence of the target interferon-αA gene with an optimized codon composition in order to provide a highly efficient translation in a plant system (Appendix A).

At the next stage of the in silico study, the consensus sequence of 5′UTR was constructed. First, we specified the criteria for selecting the size of the target regulatory sequence and its nucleotide composition, namely: (i) the size of 5′UTR characteristic of plant genes with potentially efficient translation (50 to 100 bp) based on the current standpoint (with *A. thaliana* as a model) [7,8]; (ii) the GC content meeting the average value for the 5′UTRs of plant genes; (iii) the presence of the motifs characteristic of mRNA 5′UTRs of *A. thaliana* genes (in a sample of the sequences satisfying the first two criteria); (iv) the nucleotides surrounding the start codon at position –3 (A/G) according to the Kozak sequence [19]; (v) the absence of alternative start and termination codons; (vi) the absence of secondary structure.

In total, the 5′UTR sequences of 3619 genes with a high level of transcription were selected for the analysis performed using the MUSCLE server; the multiple alignment and phylogeny of the selected sequences allowed us to group the sequences for pairwise aligning. According to the data of pairwise alignment, the motifs present in the 5′-terminal region of 50% of the sequences and having an average GC content of up to 37% were found for each group. These motifs were involved in the design of the synthetic sequence taking into account their neighborhood and localization (up- or downstream) in the *A. thaliana* 5′UTR.

The five criteria listed above were taken into account when designing the regulatory sequence as well as the secondary structure that would immediately adjoin the beginning of the start AUG codon. It is known that the 5′UTR secondary structure can have a considerable effect on translation efficiency [19]. Analysis of the synthetic regulatory sequence by the CentroidFold algorithm [20] demonstrated the absence of secondary structure (data not shown).

The performed theoretical studies allowed us to determine the 5′UTR sequence potentially able to guarantee a highly efficient translation of the target gene transcript in plant systems (Figure 1). This sequence has a size of 87 bp and GC content of 35.6%.

##### Expression Vectors for Assessing the Contribution of Optimization of Target Gene Codon Composition and 5′UTR to Heterologous Gene Translation Efficiency

Initially, we constructed the target interferon-αA gene with an optimized codon composition guaranteeing a highly efficient translation in plant systems (named IFNαA-M) via de novo synthesis of its sequence. This approach was chosen because nucleotide substitutions that were necessary in the target gene sequence are randomly localized (Appendix A). We previously convincingly showed that the insertion of IFN-αA into the loop (53 a.a.) of thermostable lichenase resulted in effective expression of the recombinant protein in *Escherichia coli* without any compromise in the biological activity of IFN-αA. Based on this, the hybrid gene LicB53-IFNαA-M was constructed; in this gene, the IFNαA-M sequence was integrated into the loop region of the gene coding for thermostable lichenase (see Section 3). Furthermore, the need for this step in the construction of hybrid genes stems from the fact that its fusion with thermostable lichenase makes it possible to precisely measure the yield of target polypeptides according to this enzyme activity with minimum time and material costs [16]. The hybrid gene LicB53-IFNαA, carrying the sequence of the native interferon-αA gene (IFNαA) integrated in the loop region of the gene coding for thermostable lichenase, was constructed earlier [16] (Figure 2).

At the next stage, we constructed the expression vectors pVIG-T-L53-IN, pVIG-T-L53-IN-M, and pVIG-T-87-L53-IN-M with the aim to find out how the modifications of the codon composition and addition of the consensus 5′UTR introduced to the target model genes according to the theoretical predictions changed their translation efficiency (Figure 1). The vectors pVIG-T-L53-IN and pVIG-T-L53-IN-M carried the hybrid gene in which the native (IFNαA) or modified (IFNαA-M) interferon-αA gene sequence were integrated in the loop region of the gene coding for thermostable lichenase, respectively. In the pVIG-T-87-L53-IN-M vector, the consensus 5′UTR deduced based on theoretical predictions (87 bp) was integrated between the CaMV 35S RNA promoter sequence and the transcription start codon (ATG) of the hybrid IFNαA-M gene (Figure 2). Note that the T-DNA region of the vector also carried the p19 gene, coding for a silencing suppressor the expression of which is controlled by the TCTP promoter and OSC terminator [21].

##### Functional Estimate of the Contribution of Target Gene Optimized Codon Composition and 5′UTR to the Expression Efficiency of Heterologous Gene

The expression efficiency of the hybrid genes carrying the native and modified interferon-αA gene sequences was assessed using transient expression. For the comparative analysis of the efficacy of genetic determinants (the codon composition of the target gene and consensus 5′UTR as regulatory translation codes), the accumulation level of the hybrid protein in the leaves of the *N. benthamiana* plants after agroinfiltration was determined. The accumulation level was computed according to the lichenase activity as part of the hybrid protein recalculated for total soluble protein (Figure 3a). The results demonstrated an increased (by 35%) content of the protein coded for by the gene with the modified codon composition as compared with the native protein, which is a considerable advantage in terms of resource expenditures.

We analyzed plant protein lysates using zymograms. On the zymogram, single cleared activity bands were detected, which may indicate that L53-IN and L53-IN-M were present in the samples in the form of monomers (Figure 4), and also that the molecular weight of the recombinant proteins L53-IN and L53-IN-M corresponds to the theoretically calculated weight of about 45 kDa, and recombinant proteins in the plant system retain the structural integrity. The biological activity of plant protein lysates L53-IN and L53-IN-M was determined using an antiviral assay (L929/VSV). As can be seen in Figure 3b, both L53-IN and L53-IN-M exhibited antiviral activity, while an increase in antiviral activity was found for the L53-IN-M protein lysate compared to that for L53-IN, which correlates with lichenase activity level (Figure 3a).

Note that quite a few studies deal with optimization of the codon composition of heterologous genes for their expression in plant systems. There are two partially overlapping directions in the optimization of the codon composition of heterologous genes, namely: (i) the replacement of synonymous triplets with those most frequently used in the genes with a high expression level; (ii) the Monte Carlo replacement method. Although several proposed software products offer broad opportunities for modification of the codon composition in target genes for their further expression in plants, they are all focused only on the maximum production of the target protein. Thus, we believe that the optimization of the target gene sequence using the algorithm we propose makes it possible to embed the expression of an additional gene into the overall array of cellular processes, affecting their function in plant cells to a lesser degree. As we see it, this approach is the most optimal since it guarantees that rare transport RNAs are not deflected from the most limiting yet important cellular processes, including those that involve genes with a high expression level. In addition to the experimental data on the use of this algorithm for optimizing the codon composition of the target model gene, interferon-αA, we tested its performance by comparing the coding sequences of model genes (native and with the codon composition optimized for efficient expression in plant systems; Appendix A–F) [21]. This in silico analysis suggests a certain universality of the proposed algorithm as well as its adequacy for optimization of the codon composition of target genes aimed at efficient protein synthesis in plant systems. It is necessary to emphasize that, although the modification of codon composition of heterologous genes is in most cases a positive component in the design of efficient expression systems for target polypeptides, the utility of its application must be nonetheless considered in each individual case.

The functional assessment of the contribution of 5′UTR to the expression efficiency of heterologous genes was performed using the expression vector pVIG-T-87-L53-IN-M. The results demonstrated that the presence of the synthetic consensus 5′UTR sequence of the heterologous gene guaranteed an increase in the accumulation of the hybrid protein by over 25% as compared with the control, as well as an increase in the antiviral activity of plant protein lysate 87-L53-IN-M compared with L53-IN-M (in this case, the expression vector pVIG-T-L53-IN-M, in which the heterologous gene did not carry 5′UTR; Figure 3). These results suggest that the synthetic consensus sequence, named 87-UTR, acted as a potential regulatory element for the translation efficiency of the heterologous gene’s mRNA. To a considerable degree, this may be determined by the presence of specific but still unknown regulatory elements in its composition; these elements may be involved in the determination of both the pattern of binding to various regulatory proteins and the stability of the transcript. In addition, an optimal neighborhood of the initiation codon in a target gene providing its efficient translation (A at position −3 and G at position +1) in the vector pVIG-T-87-L53-IN-M is most likely another positive factor in the efficient translation of target RNA in a plant system. It is known that the initiation codons in a context differing from an optimal one (AACA***ATG***GC) are less efficiently recognizable to ribosomes so that they can pass well to the next initiation codon. As is mentioned above, the synthetic consensus 5′UTR contained the most frequently occurring motifs characteristic of this region. However, it is possible to integrate into synthetic 5′UTRs not only frequently occurring motifs, but also known regulatory sequences observable in native 5′UTRs, thereby forming the basis for resolution of the relevant problems in applied plant biotechnology. Note that the presence of 5′UTR is also an important element for the ribosome positioning on the mRNA molecule; nonetheless, 5′UTR is not included in the composition of most expression vectors proposed for the expression of heterologous genes in plant systems. A series of elegant studies convincingly demonstrate the potential of heterologous 5′UTR plant genes [22], as well as some synthetic sequences [23]. They have been shown to provide high levels of heterologous gene protein product in various plant systems.

#### 2.1.2. Modulating Stability of the Protein Product: The Role of Second Amino Acid in Increasing the Target Protein Yield

As is currently believed, it is most important—even being a key task in certain cases—to preserve the already synthesized proteins, in particular, when using plants for synthesis of pharmaceutical proteins. To guarantee the stability of the protein product, it is necessary to prevent its degradation by plant proteolytic systems. Currently, a large toolkit is available and used for this purpose, including (i) expression of target genes in storage organs and tissues; (ii) intracellular compartmentalization of the protein products of target genes; (iii) co-expression of target genes with the genes coding for proteinase inhibitors; and (iv) fusion with the proteins resistant to proteolytic degradation.

Our study covered two directions: (i) stabilization of the protein product according to the N-end rule [24,25] and (ii) the use of thermostable lichenase as a PSP. The selection of these particular approaches is explainable with an insufficient degree of knowledge about the performance of the N-end rule in plants and the specific features of thermostable lichenase, which can act as a PSP [17,18].

Initially, we assessed the frequencies of individual amino acids at the second position of the overall gene pool of different plant species (Appendix A), namely, *A. thaliana*, *Zea mays*, *Solanum tuberosum*, *Hordeum vulgare*, and *Oryza sativa*. The distribution of the genes carrying a particular amino acid at the second position for the selected plant species was rather conserved (Appendix A). As a consequence, the distributions of the genes carrying at the second position either a stabilizing amino acid residue or a destabilizing amino acid residue were similar in different plant species. According to the N-end rule, the protein half-life is determined by a particular N-terminal amino acid residue of its polypeptide chain; note that the stabilizing residues were represented by six amino acids: Met, Gly, Val, Thr, Ser, and Ala. The available experimental data demonstrate that Gly, Val, and Ala at the second position influenced protein stability to a similar degree but less than the Thr and Ser residues [24,25]. Note that the Thr and Ser residues are coded for by codons with a first nucleotide other than G, whereas Gly, Val, and Ala have G at the first position. This fact is important with respect to the preservation of the Kozak consensus sequence (the neighborhood of the start AUG codon, in particular, the prevalence of guanine (G) at position +4). Note that the frequency of Gly, Val, and Ala as the second amino acids according to their distribution in plant genes was Ala > Gly > Val (Appendix A). Correspondingly, the selection of Ala or Gly as a stabilizing amino acid residue was preferable. When choosing the destabilizing amino acid residue for our experiments, we relied on the most pronounced destabilizing effect of an amino acid residue. The relevant published data convincingly demonstrate that the destabilizing amino acid residues of the first order—Arg, Lys, His, Phe, Trp, and Tyr—have the most pronounced effect [24,25]. Note that none of these amino acids is coded for by a codon with G as the first nucleotide; moreover, these amino acid residues have similar effects on protein instability. The frequencies of these amino acids at the second position according to their distribution in plant genes is Arg > Lys > His > Phe > Trp > Tyr. This led us to select Arg as the destabilizing amino acid residue.

##### Expression Vectors for Assessing the Contribution of the Second Amino Acid to the Increase in the Yield of Protein Product of Heterologous Genes

The degree of the effect of a destabilizing or a stabilizing amino acid residue at the second position on the stability of the protein product in plant cells was clarified in special experiments. Initially, we constructed the vectors for transient expression in plants with the triplets coding for either stabilizing or destabilizing amino acids as the regulatory determinants of stability, taking into account that these triplets must be localized immediately downstream of the start codon. The vector pVIG-T-L53-IN-M carried the triplet coding for glycine (this is the second amino acid residue of the thermostable lichenase N-terminal region) as the regulatory determinant of stability. Using molecular cloning procedures, we constructed the vector pVIG-T-Arg-L53-IN-M, in which the triplet coding for Gly, a stabilizing amino acid, was replaced with Arg, a destabilizing amino acid (as the second amino acid residue in the thermostable lichenase N-terminal region; Figure 2).

##### Functional Assessment of the Contribution of the Second Amino Acid to the Increase in the Yield of Protein Product of Heterologous Genes

The functional assessment of the contribution of the second amino acid residue to the stability of the protein product of a heterologous gene showed that the replacement of the triplet coding for a stabilizing amino acid residue at the second position with the codon corresponding to a destabilizing amino acid in the expression vector decreased the yield of the hybrid protein by 35% (Figure 3a). Note that the replacement of the second amino acid in the recombinant protein with a destabilizing residue also led to a decrease in the antiviral activity of the recombinant Arg-L53-IN-M protein compared to L53-IN-M in a similar manner (Figure 3b). These data agree with the earlier results. In particular, the use of a stabilizing residue at the second position has been demonstrated to cause a 20% increase in the yield of reporter protein as compared with the control carrying methionine as the second amino acid residue versus the destabilizing residue, decreasing the yield by 80% [24]. However, note that here, we speak about the experiment in which ubiquitin and the target protein were in equimolar concentrations because they were fused in one expression cassette. In our study, the stability of the hybrid polypeptide was determined by the ubiquitin system of the plant cell alone, which, as we see it, explains certain differences in the observed quantitative yields of the protein product; in addition, another possible cause of certain discrepancies is the used reporter proteins. Undoubtedly, a reasonable question arises on what the particular factor is that causes a decrease in the yield of the target protein, namely, whether it is a nonoptimal neighborhood of the initiation codon when using the triplet coding for Arg or the contribution of the Arg residue itself. It was experimentally demonstrated for several proteins with a stabilizing amino acid at the second position coded for by a triplet with a first nucleotide other than G that the protein product was synthesized at a sufficiently high yield (exceeding the corresponding control). For example, this has been shown for threonine, a stabilizing amino acid, which is coded for by the triplet lacking G at the first position. This suggests that an increase in the stability of the target protein in our case is a considerable contributor to its yield. As is mentioned above, the differences in the yield of recombinant protein when using stabilizing and destabilizing amino acid residues reached 35–80%. We believe that this approach to the provision of stability of the target polypeptides is potentially effective because the presence of the triplets coding for a stabilizing amino acid residue directly in the expression vectors can be preplanned [21]. In addition, the use of the codon for glycine in this role allows the Kozak consensus sequence (providing the recognition of the triplet coding for the first amino acid residue of the polypeptide chain by the ribosome) to be integrated in the neighborhood of the start codon, that is, the additional genetic determinants guaranteeing an efficient expression of target genes in plants at both the translation level (Kozak sequence) and the stage of protein stability preservation (second amino acid) to be added to the plant vector.

#### 2.1.3. Modulating Stability of Protein Product: The Role of PSP in the Increase in Target Protein Yield

In our opinion, the second promising approach to increase the stability of the protein product of target genes is the fusion of the target polypeptide with a stable protein. Correspondingly, we tested the *C. thermocellum* thermostable lichenase as a PSP for plant expression systems. The thermostable lichenase has previously been successfully tested as a reporter protein and a carrier protein for target proteins in prokaryotic cells [16]. However, this enzyme has not been adequately considered as a potential PSP for plant systems. The main properties of this protein that are important in this respect are its high specific activity, thermostability, lower susceptibility to the hydrolysis by proteolytic enzymes, preservation of specific activity after fill-ins at the regions of amino acid residues 53 and 99, and functional folding of the protein domains (GFP or interferon) integrated into thermostable lichenase [16], at least in prokaryotic expression systems.

##### Expression Vectors and Functional Assessment of the Contribution of Thermostable Lichenase as a PSP to the Increase in the Yield of the Protein Product of Heterologous Gene

The transient expression of the native IFNαA-M gene and the hybrid gene of interferon-αA as an internal module within LicBM3 was compared to find out the performance of thermostable lichenase as a PSP when producing the target protein in a plant system. For this purpose, we constructed an additional vector, pVIG-T-IN-M (Figure 2). Next, the *N. benthamiana* tobacco leaves were agroinfiltrated using two expression vectors: (i) pVIG-T-L53-IN-M, carrying the hybrid gene (L53-IN-M) with the modified sequence of interferon-αA gene (IFNαA-M) integrated into the loop region of the gene coding for thermostable lichenase, and (ii) pVIG-T-IN-M, carrying the native gene (IFNαA-M) without the integration into the thermostable lichenase gene sequence. In order to assess the accumulation level of the native interferon-αA (IFNαA-M) and the hybrid protein (L53-IN-M) in plant systems, we first compared the ability of plant protein lysates containing either the native or hybrid interferon to inhibit the cytopathic effect of the test virus, VSV, in the L929 continuous cell culture. Note that all tested proteins were expressed and purified in an identical manner (see Materials and Methods) to exclude any potential differences in their activity resulting from the use of different expression systems or purification techniques. As was expected, the plant protein lysates (TPL) carrying thermostable lichenase LicB53 displayed no antiviral activity (data not shown) versus the samples of L53-IN-M, with an antiviral activity (CPE_50_) of 15.25 ± 0.21 ng/mL, and the samples of IFNαA, with an antiviral activity of 21.55 ± 0.34 ng/mL (Figure 3b). Thus, the comparative analysis demonstrates that the accumulation level of native interferon-αA in the plant system was 40% lower as compared with that of the hybrid protein in which interferon-αA was fused with thermostable lichenase.

These results demonstrate that the PSP involving thermostable lichenase not only failed to cause any significant negative effect on the activity of the target IFNαA protein but rather increased the activation of the target protein in the plant cell. We previously showed that the insertion of interferon-αA into the loop (53 aa) of *C. thermocellum* thermostable lichenase results in an efficient expression of the soluble recombinant protein in *E. coli* periplasm without compromising the interferon-αA biological activity [16].

The computations based on the experimental data convincingly demonstrate a statistically significant increase in the level of the L53-IN-M hybrid protein in plant cells, which is most likely determined by the specific features in the folding of thermostable lichenase, the catalytic domain of which forms a classical β-jellyroll fold; in general, this fold is amazingly stable with respect to conformation retention, resisting changes in both amino acid sequence and chain topologies [16]. This suggests a reasonable inference that the conformational stability of the recombinant protein L53-IN-M is able to determine a more efficient expression of this protein in plant cells as compared with IFNαA-M. According to current opinion, proteins with an enhanced stability are tolerant to mutational events, including proteins with a terminal fusion or insertion of domain(s), and retain their functionality [16], as we previously demonstrated for thermostable lichenase [16].

##### Possible Purification of Recombinant Protein Using Thermostable Lichenase as a PSP

We previously demonstrated the in vitro refolding of lichenase and target proteins integrated into the lichenase loop domains by exposing protein lysates to 65 °C or precipitating with ethanol [16]. Correspondingly, we decided to experimentally find out (i) whether the hybrid protein (L53-IN-M) was able to in vitro refold after the incubation of plant protein lysates at 65 °C and subsequent ethanol denaturation; (ii) how the amount of total soluble proteins decreased in plant lysates after these procedures; and (iii) whether the target protein, interferon-αA (IFNαA-M), retained its biological activity. The refolding of the hybrid protein, lichenase (L53-IN-M) with inserted interferon-αA (IFNαA-M), was assessed according to the restoration of the enzyme (lichenase) activity and the ability of the hybrid protein to inhibit the test virus CPE in the L929 continuous cell culture [16].

It was shown that the lichenase activity of both the initial variant (LicBM) and the protein variant with inserted domain not only retained but even increased in a statistically significant manner (at least tenfold; Figure 5), while there was a decrease in total soluble proteins in plant lysates (Appendix A) and an increase in the amount of recombinant protein L53-IN-M in plant protein lysates after heating and treatment with ethanol (Figure 6). This increase in the enzyme activity is explainable with a considerable decrease in the share of the other total soluble proteins (10- to 20-fold) in the protein samples after ethanol precipitation (Figure 4). Note that the inserted domain (IFNαA-M) in L53-IN-M was also able to refold in vitro after heating and ethanol denaturation, as evidenced by the results of the antiviral activity of the plant protein lysates before and after heating and precipitation with ethanol (Figure 5). On the zymogram, single cleared activity bands were detected in the plant protein lysate of L53-IN-M after ethanol denaturation (Figure 4). The preservation of lichenase activity and biological activity of interferon after the incubation of plant protein lysates at 65 °C and precipitation of nontarget proteins with ethanol can be used for the rapid and efficient purification of fusion proteins (Figure 5).

It should be emphasized that plants contain a tremendous number of secondary metabolites, which are potentially useful for many purposes, in particular, as anticancer drugs. These metabolites are extractable from plants concurrently with the purification of proteins fused with thermostable lichenase. Currently, there are two main options for isolating such metabolites from the side flows while manufacturing recombinant proteins, namely, (i) purification from the primary extract after target protein capture, and (ii) re-extraction from the residual biomass using solvents with orthogonal solubilizing properties as compared with the primary buffer, i.e., organic liquids such as methanol or ethanol [26,27]. For example, tobacco plants, frequently used for manufacturing plant-made pharmaceuticals [28], contain several different flavonoids and alkaloids, first and foremost, nicotine [29], which are moderately soluble in aqueous solutions commonly used for protein extraction [30]. However, these small molecules can be released from plant biomass via re-extraction with alcohols [31]. Of the tobacco metabolites, rutin is potentially applicable in breast cancer treatment [32].

## 3. Materials and Methods

### 3.1. Methods for In Silico Analysis

Identification of the potential regulatory contexts in the noncoding mRNA regions, the search for consensuses predicted based on 5′UTR selection, the optimization of the codon composition of heterologous genes, and the analysis of the codons for the amino acid at the second position were performed using the public JetGene database and the corresponding software (https://jetgene.bioset.org/, accessed on 1 January 2022) [33]. This resource contains CDSs and cDNA, 5′UTR, and 3′UTR sequences of six major groups of living organisms, plants included. The JetGene software makes it possible to form samples of nucleotide coding and regulatory sequences and further analyze these sequences according to various parameters using the designed algorithms. *Arabidopsis thaliana* gene coding sequences and the CDS Data section (Codon Usage subsection) were used to optimize the codon composition of the target gene; this allowed the correspondence of the codon composition of the target gene to be assessed and the target gene sequence to be optimized via in silico replacement of codons. The synthetic 5′UTR sequence was designed using the 5′UTR Data section (GC-Content in 5′UTR and Motifs subsections), and the codon at the second position was selected using the CDS Data section (Codon Position subsection). The MUSCLE program was used to align nucleotide sequences [34].

### 3.2. Engineering Plant Expression Vectors

The vector pVIG-T [21] was used to construct the following vectors for transient expression: pVIG-T-L53-IN, pVIG-T-L53-IN-M, pVIG-T-87-L53-IN-M, pVIG-T-Arg-L53-IN-M, and pVIG-T-IN-M. Construction of the vectors comprised the following stages (see Appendix A for the primers used):Construction of the modified lichenase gene (L53) carrying two *Bsp*QI sites integrated at position 53 aa; these sites form the cloning region of the target inserts according to the GoldenGate technology. The pQE30-*lic*BM3 plasmid was linearized using the L53_F1/L53_R1 and L53_F2/L53_R2 primer pairs; the resulting amplicons were joined and hybridized. The synthetic insert (L53dummy_S/L53dummy_AS) was integrated using a SLIC [35] approach to give the plasmid pQE30-L53.Integration of the modified thermostable lichenase gene into the intermediate pFGG plasmid. The pFGG vector was linearized by digestion at *Apa*I/*Sma*I sites to clone the modified thermostable lichenase gene *lic*BM3-53 (L53) using the L_Apa_F/L_Sma_R primer pair from the pQE30 plasmid; the sticky ends were formed with *Apa*I and *Sma*I restriction endonucleases; thus, the pFGG-L53 vector was constructed.Construction of the variants of modified lichenase carrying destabilizing amino acid at the second position (Arg-L53) or synthetic 5′UTR (87-L-53). The pFGG-L53 plasmid was linearized by PCR with the L53_ARG_F/L53_ARG_R primer pair. Both primers carried an additional triplet coding for arginine, the destabilizing amino acid. The linearized plasmid was ligated at the blunt ends to form the circular plasmid pFGG-Arg-L53. The pFGG-L53 plasmid was linearized by PCR with the L53_87_F1/L53_87_R1 and L53_87_F2/L53_87_R2 primer pairs. The synthetic insert corresponding to the 5′UTR (UTR87S/UTR87AS) was integrated using a SLIC approach to obtain the final vector pFGG-87-L53.Cloning of the modified and native interferon genes to position 53 aa. The native and modified interferon genes were amplified using the INN_F1/INN_R1 and INN_F2/INN_R2 primer pairs for the native variant and INM_F1/INM_R1 and INM_F2/INM_R2 for the modified one. The resulting products were joined and hybridized to obtain sticky ends. The initial pFGG-Arg-L53 plasmid was linearized by hydrolysis with BspQI restriction endonuclease. The vector and the insert were ligated using a standard procedure (pFGG-L53-INM, pFGG-L53-87-INM, and pFGG-L53-Arg-INM). To integrate the native interferon gene, the amplicon produced by PCR with the INM_Apa_F/INM_Sma_R primer pair was hydrolyzed at the *Apa*I/*Sma*I sites and further cloned into the pFGG vector similar to different genes of thermostable lichenase.Transfer of the produced expression cassettes carrying hybrid genes derived from the nucleotide sequences of lichenase and interferon genes to the vector optimized for transient expression of heterologous genes in plants, pVIG-T. All expression cassettes were transferred from the vectors pFGG-L53-IN, pFGG-L53-IN-M, pFGG-87-L53-IN-M, pFGG-Arg-L53-IN-M, and pFGG-IN-M to the vectors pVIG-T (plasmids pVIG-T-L53-IN, pVIG-T-L53-IN-M, pVIG-T-87-L53-IN-M, pVIG-T-Arg-L53-IN-M, and pVIG-T-IN-M were formed) using a routine restriction–ligation cloning technique at *Spe*I/*Hpa*I sites (Appendix A).

### 3.3. Plant Material, Agrobacterium Strains, and Agroinfiltration

*N. benthamiana* plants were hydroponically grown at a temperature of 22 ± 2 °C, with a photoperiod of 8 h, and at an illumination of 100 μmol quanta/(m^2^ s). Knop’s solution was used as the nutrient medium. The pVIG-T-L53-IN, pVIG-T-L53-IN-M, pVIG-T-87-L53-IN-M, pVIG-T-Arg-L53-IN-M, and pVIG-T-IN-M vectors were used to transform (by electroporation) the *A. tumefaciens* GV3101 strain as previously described [36]. The transformed bacteria were selected on a medium containing kanamycin and named according to the expression vector used for transformation. The transformed agrobacteria were used for agroinfiltration into the abaxial epidermis of the leaves of 6-week-old *N. benthamiana* with a syringe without a needle as previously described [37]. After agroinfiltration, the plants were cultivated under the same conditions for 7 days. Agroinfiltrated leaf fragments were used in subsequent analysis.

### 3.4. Producing Plant Protein Extracts

For extraction of the total soluble protein, plant tissue frozen in liquid nitrogen was ground in 0.05 M acetate buffer pH 6.0 (2 mL of the buffer per 1 g of plant tissue) and centrifuged at 10,000× *g* for 10 min; the supernatant was collected for further experiments. This sample is referred to as total protein lysate (TPL). Part of the TPL (1.5 mL) was incubated at 65 °C for 60 min followed by centrifugation at 10,000× *g* for 10 min. The insoluble fraction was separated by centrifugation, and the supernatant was used for further analyses and named PLTTs (protein lysates after temperature treatment). Part of the PLTT (0, 75 mL) was treated with three volumes of 96% ethanol at 4 °C for 8–10 h followed by centrifugation at 10,000× *g* for 10 min. The sediment was dried, dissolved in 0, 75 mL of 50 mM Tris–HCl (pH 8.0), and used as preparations of proteins after ethanol precipitation (PEPs). Protein concentration was determined according to Bradford using bovine serum albumin as a standard [38].

### 3.5. Biochemical Methods for Determining Lichenase Activity and Interferon Biological Activity

The lichenase activity displayed by native and hybrid proteins was determined by fluorescence technique using lichenan at a concentration of 125 mg/mL as the substrate and Congo Red (CR) at a final concentration of 0.005% as the dye. The fluorescence intensity was assessed in a Synergy H1 (BioTek, Santa Clara, CA, USA) multimode photometer using 96-well microtiter plates at λ_Ex_ = 550 nm and λ_Em_ = 620 nm. The blank samples containing (i) CR (dye background) and (ii) lichenan (125 µg/mL) + CR (substrate background) were used in all measurements. For quantitative estimation, the values of fluorescence intensity were calculated net of the dye and substrate backgrounds using numerical analysis [39].

The activities of native (IN-M) and hybrid (L53-IN-M) interferons and thermostable lichenase (LicB-53; control) were determined according to the inhibition of the cytopathic effect of a test virus (vesicular stomatitis virus, VSV, strain Indiana at a dose of 100 CPE_50_) in the L929 continuous cell culture grown in 96-well Linbro culture plates. The value inverse to the dilution of the sample providing a 50% cell protection from the CPE of the test virus was regarded as the titer of the protein. Briefly, the L929 cells (Merck KGaA, Darmstadt, Germany) were diluted with the medium supplemented with 10% fetal bovine serum (FBS) to 2.5 × 10^5^–3.5 × 10^5^ cells/mL. Each well of a 96-well plate was inoculated with diluted cells (100 µL). Samples of plant protein lysates diluted with the medium containing 10% FBS to the necessary concentrations according to the total soluble protein (0.95, 1.9, 3.81, 7.62, 15.25, and 30.5 ng/mL) were incubated with L929 cells for 16–18 h at 37 °C. The medium was then replaced with DMEM containing 7% FBS and 100 CPE_50_ VSV. After 24 h of incubation, cell viability was assessed using an MTT assay [40]. Optical density (OD) was measured in a PHERAstar FSX (BMG Labtech, Ortenberg, Germany) multimodal reader. The amount of total soluble protein obtained from the plant material and required for 50% cell protection from the test virus CPE was used to assess the accumulation level of the native (IN-M) and hybrid (L53-IN-M) interferons in the plant system. The experiments were performed in triplicate. Note that the CPE_50_ values reported by different laboratories may differ because of different experimental conditions [16].

The protein preparations were subjected to electrophoresis in polyacrylamide gel under denaturing conditions according to the Laemmli method [41].

Zymograms were obtained by staining the gel after protein separation by SDS-PAGE (12%) in the presence of 0.1% lichenan as described earlier [16]. Molecular weights were estimated using a Prestained Protein Ladder (Fermentas, Vilnius, Lithuania). After electrophoresis, the gels were incubated at 70 °C for 1 h. Enzyme activities were determined by staining with 0.5% Congo Red solution followed by washing in 1 M NaCl.

Western blot analysis was performed with antihumIFN α goat antiserum (ab193823; Abcam, Cambridge, UK) as primary antibodies and rabbit anti-goat HRP (ab6741; Abcam) as secondary antibodies. After washing, each PVDF membrane (Bio Rad, Berkeley, CA, USA) was stained by electro-chemiluminescent method (ECL) (GE Healthcare, Chicago, IL, USA). Chemiluminescence was detected using a Fusion FX5 device (VilberLourmat, Collégien, France).

### 3.6. Statistical Data Processing

All experiments were performed in at least eight–ten independent replicates. The data were processed using Statistica for Windows v. 9.0 and Microsoft Office Excel 2007 unless otherwise specified. The experimental measurements were made in five–ten analytical replicates. Arithmetic means and their standard errors are shown in the figures and tables unless otherwise specified.

## 4. Conclusions

A broad and amazingly intricate network of mechanisms underlying the decoding of a plant genome into the proteome forces the researcher to design new strategies to enhance both the accumulation of recombinant proteins and their purification from plants and to improve the available relevant strategies. In this paper, we searched for new approaches to modulate the efficiency of key cellular processes aiming to increase the yield of the target protein in a plant system and tested their performance. Our data convincingly demonstrate that the use of an integrated approach comprising an in silico analysis and experimental verification of the proposed hypotheses is the most optimal for resolving a number of problems. We designed new approaches to optimize the codon composition of target genes (case study of interferon-αA) and to search for regulatory sequences (case study of 5′UTR) with the aim to modulate the cellular process of translation and, as a consequence, to achieve efficient synthesis of the protein products of target genes in plant systems. In addition, we have convincingly shown that the approach utilizing stabilization of the protein product according to the N-end rule or a new protein-stabilizing partner (thermostable lichenase) is sufficiently effective and results in a significant increase in the protein yield manufactured in a plant system. Moreover, it was validly demonstrated that thermostable lichenase as a protein-stabilizing partner not only has no negative effect on the target protein activity (in our case, IFNαA) integrated in its sequence, but rather enhances the accumulation of the target protein product in plant cells. In addition, the retention of lichenase enzyme activity and interferon biological activity after the incubation of plant protein lysates at 65 °C and precipitation of nontarget proteins with ethanol suggests that this is applicable to the rapid and efficient purification of fusion proteins, thereby confirming the utility of thermostable lichenase as a protein-stabilizing partner for plant systems.

Thus, our experimental data validated the advantage of the studied regulatory codes and protein-stabilizing partner in considerably increasing the yield of the target protein in plant systems and its purification, which, depending on the scale of protein synthesis, can provide a significant benefit in the perspective of resource expenditures.

## Figures and Tables

**Figure 1 plants-11-02450-f001:**
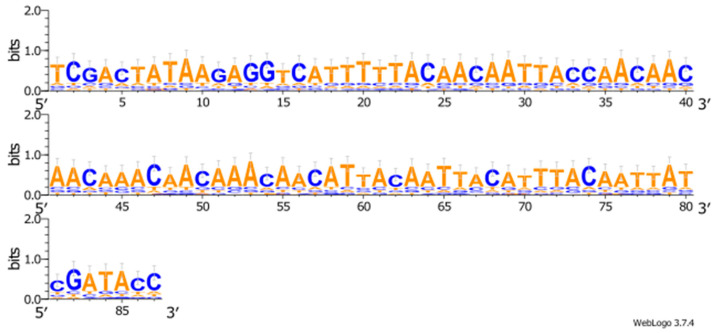
Consensus 5′UTR deduced based on theoretical predictions (87 bp).

**Figure 2 plants-11-02450-f002:**
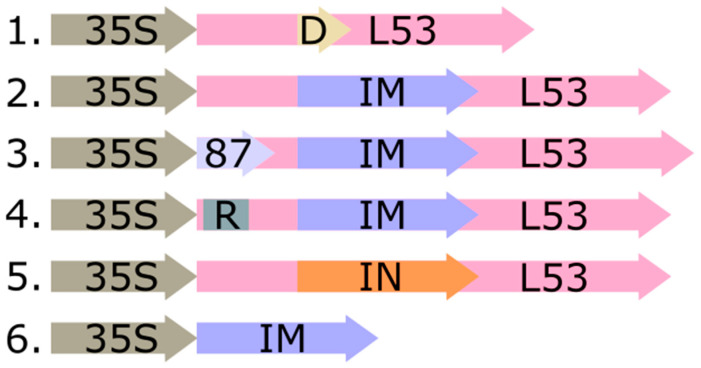
Scheme of the target part of T-DNA expression vectors pFGG-L53 (1), pVIG-T-L53-IN (2), pVIG-T-L53-IN-M (3), pVIG-T-87-L53-IN-M (4), pVIG-T-Arg-L53-IN-M (5), and pVIG-T-IN-M (6); 35S—35S RNA CaMV promoter; L53—sequence of thermostable lichenase reporter gene; D—cloning area of target inserts using GoldenGate technology; IM—modified interferon-αA gene (IFNαA-M); 87—consensus 5′UTR deduced based on theoretical predictions (87 bp); R—Arg, a destabilizing amino acid (as the second amino acid residue in the thermostable lichenase N-terminal region); IN— native interferon-αA gene (IFNαA).

**Figure 3 plants-11-02450-f003:**
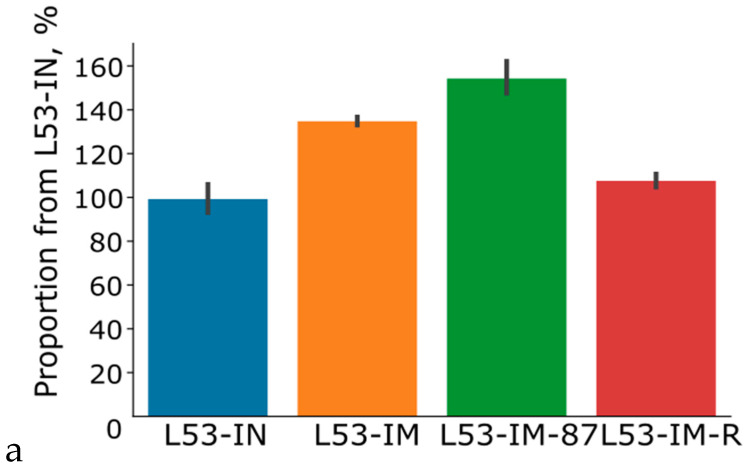
Comparative analysis of the activity of thermostable lichenase (**a**) and biological activity of interferon-αA (**b**) as parts of a recombinant protein using different expression vectors for transient expression. **L53-IN**—transient expression using the vector pVIG-T-L53-IN to carry out the sequence of the native interferon-αA gene (IFNαA) integrated in the loop region of the gene coding for thermostable lichenase; **L53-IM**—transient expression using the vector pVIG-T-L53-IN-M to carry out the sequence of the modified interferon-αA gene (IFNαA-M) integrated in the loop region of the gene coding for thermostable lichenase; **L53-IM-87**—transient expression using the pVIG-T-87-L53-IN-M to carry out the consensus 5′UTR deduced based on theoretical predictions (87 bp) integrated between the CaMV 35S RNA promoter sequence and the transcription start codon (ATG) of the hybrid IFNαA-M gene; **L53-IM-R**—transient expression using vector pVIG-T-Arg-L53-IN-M to carry out the sequence of the modified interferon-αA gene (IFNαA-M) integrated in the loop region of the gene coding for thermostable lichenase in which the triplet coding for Gly, a stabilizing amino acid, was replaced with Arg, a destabilizing amino acid (as the second amino acid residue in the thermostable lichenase N-terminal region).

**Figure 4 plants-11-02450-f004:**
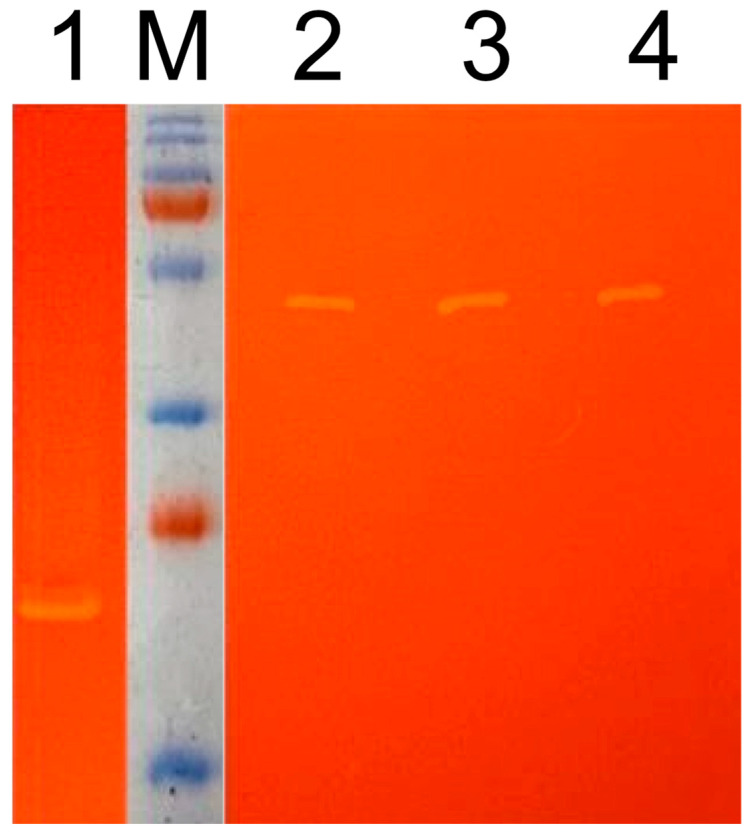
Zymogram of plant protein lysates. 1—LicBM3, native variant of thermostable lichenase (27 kDa); 2—L53-IN, transient expression using the vector pVIG-T-L53-IN to carry out the sequence of the native interferon-αA gene (IFNαA) integrated in the loop region of the gene coding for thermostable lichenase; 3—L53-IM, transient expression using the vector pVIG-T-L53-IN-M to carry out the sequence of the modified interferon-αA gene (IFNαA-M) integrated in the loop region of the gene coding for thermostable lichenase; 4—L53-IN-M after ethanol denaturation; M—molecular weight marker.

**Figure 5 plants-11-02450-f005:**
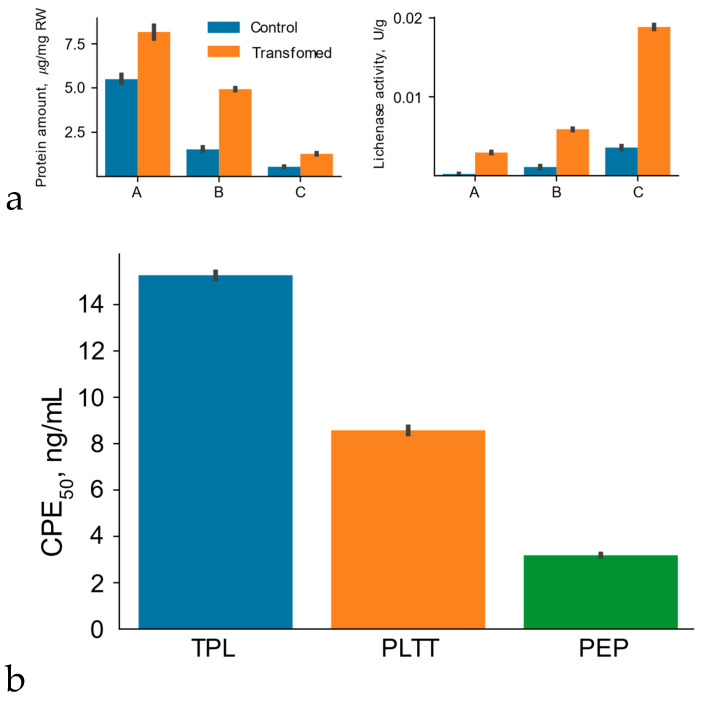
Total soluble protein (**1**) and lichenase activity (**2**) of plant protein lysates before (A) and after incubation at 65 °C (B) and subsequent ethanol denaturation (C) (**a**). Control—control *N. benthamiana* plants (without agroinfiltration); Transformed—*N. benthamiana* plants after agroinfiltration using the vector pVIG-T-L53-IN-M; µg/mg RW—the amount of total soluble protein in terms of 1 μg of fresh weight of plant tissue. Biological activity of interferon-αA (**b**). TPL—total protein lysate after transient expression using the vector pVIG-T-L53-IN-M; PLTT—the protein lysates after temperature treatment; PEP—the protein lysate after temperature treatment and ethanol precipitation. Samples balanced for the amount of total soluble protein are applied.

**Figure 6 plants-11-02450-f006:**
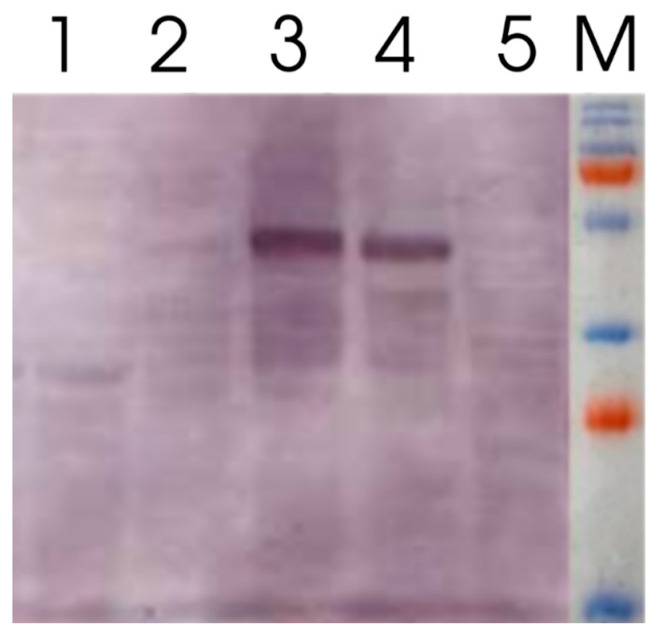
Western blot hybridization of total plant protein lysates. 1—native interferon αA (IFNαA); 2—total protein lysate (TPL) after transient expression using the vector pVIG-T-L53-IN-M; 3—the protein lysate after temperature treatment and ethanol precipitation (PEPs); 4—the protein lysates after temperature treatment (PLTT); M—molecular weight marker. Samples balanced for the amount of total soluble protein are applied to each lane.

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
