# Peer review of "Modulation of the Translation Efficiency of Heterologous mRNA and Target Protein Stability in a Plant System: The Case Study of Interferon-αA"

_plants, 2022, doi:10.3390/plants11192450_

Round 1

Reviewer 1 Report

This manuscript investigates the strategies of recombinant protein production in tobacco plants, using the interferon alpha as an example. The authors explored the strategies to increase the protein yield while maintains stability. The authors have demonstrated that using a 5’-UTR for increased protein production, the designed second amino acid sequence and L53 loop insertion for protein stabilization. It is beneficial to the general audience that these areas have been carefully explored and the data are presented. This will provide information for the scholars in the field using the plant system for recombinant protein manufacturing. Therefore, the manuscript has the significance of advancing the field. However, the manuscript can be improved according to the following suggestions:

1)      Remove excessive and extrapolated comments and discussions that are not backed up by your results or the literature. For example: Page 6, line 261-267; Page 7, line 308-314, etc.

2)      Page 4, line 185-187: I am not quite sure what the authors mean here. The entire paper has not mentioned anything about secondary structure anywhere else.

3)      It is critical to run SDS-PAGE and immunoblot before and after the heat treatment to assess the size and the integrity of the recombinant protein target.

4)      The authors listed the interferon biological activity, but the authors did not present the data. These data are important to support the conclusion that inserting into the lichenase loop does not interfere with its activity.

5)      Figure 2: the authors include every symbols in the legend but “D” in construct 1.

Author Response

Dear Reviewer:

Thank you for your letter and for editor’s and reviewer’s comments concerning our manuscript. Those comments are all valuable and very helpful for revising and improving our paper, as well as the important guiding significance to our research. We have studied comments carefully and have modified the manuscript.

The main corrections in the paper and the responds to the reviewer’s comments are as flowing:

Reviewer#

RESPONSES TO COMMENTS

  1. Comment: Remove excessive and extrapolated comments and discussions that are not backed up by your results or the literature. For example: Page 6, line 261-267; Page 7, line 308-314, etc.?
  • done.

  1. Comment: Page 4, line 185-187: I am not quite sure what the authors mean here. The entire paper has not mentioned anything about secondary structure anywhere else.
  • Thanks for the comment, an indication of the importance of the secondary structure of the 5'-UTR was overlooked. Added paragraph to text.

  1. Comment: It is critical to run SDS-PAGE and immunoblot before and after the heat treatment to assess the size and the integrity of the recombinant protein target.
  • Figures 4 and 6 as well as and their description have been added into the manuscript.

  1. Comment: The authors listed the interferon biological activity, but the authors did not present the data. These data are important to support the conclusion that inserting into the lichenase loop does not interfere with its activity.
  • Figures 3b and 5b as well as and their description have been added into the manuscript.

  1. Comment: Figure 2: the authors include every symbols in the legend but “D” in construct 1.
  • Done.

We tried our best to improve the manuscript and made some changes in the manuscript, and these changes will not influence the content and framework of the paper.

We appreciate for Reviewer’ warm work earnestly, and hope that the correction will meet with approval. 

Once again, thank you very much for your comments and suggestions.

Best regards.

Yours sincerely,

Reviewer 2 Report

The submission is about the increase of foreign protein expression by modulation of the 5’ UTR sequences (in silica and validations), codon optimization based on the recipient plant, and stabilization of protein according to the N-end rule or by fusion with a new protein stabilizing partner (PSP) thermostable lichenase. A transient system was used to validate the various improvements on the alfa-interferon, for which expression is indirectly measured by the lichenase enzymatic activity. Maintenance of interferon activity is also compared between the interferon and interferon-lichenase fusion protein expression constructs.

It is an ambitious project but unfortunately not well supported by experimental data. There are various improvements needed before publication.

The major issues are:

1.       Problem for 5’UTR design (lane175), a total number of 3619 arabidopsis genes 5’UTR sequences were used for designing optimal 5’UTR, this practice has two problems:

5a) No consideration of downstream gene expression level (temporal/spatial expression level). A more appropriate practice might be the use of the top 100 genes that are ubiquitously expressed in the vegetative tissues.

5b) Also, tobacco genome sequences might be more appropriate for such 5’UTR optimization for protein expression in N. Benethamiana.

Authors need to justify this practice in reference to the other possibilities.

2.       It is noted that instead of testing interferon protein quantity and bioactivity, lichenase enzymatic activity was used as the representation of interferon for four constructs (Figure 3). This is not appropriate. Not only because it is indirect, but also the lack of a close relationship between alpha-interferon (quantity and bioactivity) with the lichenase enzymatic activity. Interferon protein concentration needs to be provided, its bioactivity should be there too.

3.       Alpha-interferon is a pharmaceutically active protein. Any novel expression system of such protein needs evidence of the maintenance of its activity. Authors presented antiviral activity data (CPE50) (lanes 435-436) and concluded “accumulation level of native interferon-aA.. is by 40% lower as compared with that for the hybrid protein…”. Two questions:

2a. It is not clear which protein preparation was used for this bioassay: TPL, PLTT or PEP?

2b. The suggestion that the increase of bioactivity is due to a higher amount of protein may or may not be true, needs some more evidence. The alternative possibility of better protein stability needs to be evaluated.   

4.       If expressed as a fusion protein, a strategy for purification of the protein is necessary for pharmaceutical use, otherwise there is the need for justification for using the fusion protein directly for treatment.  

5.       Figure 4 intends to showcase value of heat treatment or ethanol precipitation for purification of protein but needs some improvements (lanes 481-486):

Its two panels give total soluble protein (panel 1 on the left) and lichenase activity (panel 2 on the right) for A: total protein lysate (TPL); B: protein lysate after temperature treatment (PLTT) after 65oC incubation, proteins after ethanol precipitation (PEP). Blue columns are for control N. benethamiana plants and yellow columns are for the pVIG-T-L53-In-M transformed plants (in reference to its M&M part, lanes 575-586). It is noted that there is no volume data on dissolving ethanol precipitated protein.

5a. panel 1 suggests that transformed plants have higher total protein concentration, the increase of protein mostly remain soluble after heat treatment. Since the data are represented by concentration (ug/ml), the buffer volume to redissolve ethanol precipitate should be given for better interpretation. A better presentation is the use of protein quantity per fresh weight of leave (ug / mg fresh weight).

5b. Panel 2, on the other hand, seems to suggest that specific activity for lichenase increased after heat treatment; such increase is even more so (>10x) after ethanol precipitation. Authors suggest (lanes 473-476) that “a considerable decrease in the share of the other total soluble proteins”. This is not convincing enough. Authors need to provide some protein purity data, preferably SDS-PAGE gel staining to show increased protein purity. It is also suggested to show changes in total protein quantity.

Author Response

Dear Reviewer:

Thank you for your letter and for editor’s and reviewer’s comments concerning our manuscript. Those comments are all valuable and very helpful for revising and improving our paper, as well as the important guiding significance to our research. We have studied comments carefully and have modified the manuscript.

The main corrections in the paper and the responds to the reviewer’s comments are as flowing:

Reviewer##

RESPONSES TO COMMENTS

  1. Comment: 1Problem for 5’UTR design (lane175), a total number of 3619 arabidopsis genes 5’UTR sequences were used for designing optimal 5’UTR, this practice has two problems:

1a) No consideration of downstream gene expression level (temporal/spatial expression level). A more appropriate practice might be the use of the top 100 genes that are ubiquitously expressed in the vegetative tissues.

  • The level of translation has been little studied for plants, so we relied on the level of transcription. Was omitted from the description, added to the text of the manuscript.

1b) Also, tobacco genome sequences might be more appropriate for such 5’UTR optimization for protein expression in N. benethamiana. Authors need to justify this practice in reference to the other possibilities.

  • Currently, the tobacco genome is poorly annotated, making it difficult to find effective regulatory sequences. In addition, there are examples of the effective use of heterologous regulatory sequences, including viral and synthetic ones. Relevant information has been added to the text of the manuscript.

.

  1. Comment: It is noted that instead of testing interferon protein quantity and bioactivity, lichenase enzymatic activity was used as the representation of interferon for four constructs (Figure 3). This is not appropriate. Not only because it is indirect, but also the lack of a close relationship between alpha-interferon (quantity and bioactivity) with the lichenase enzymatic activity. Interferon protein concentration needs to be provided, its bioactivity should be there too.
  • Enzymogram (Figure 4) and Western blot data (Figure 6) were added to assess the size and integrity of the recombinant target protein, as well as data on the biological activity of interferon. The amount of interferon correlates with the amount of recombinant protein, which includes interferon, as we have previously demonstrated. Corrections and additions have been made to the text of the manuscript.

  1. Alpha-interferon is a pharmaceutically active protein. Any novel expression system of such protein needs evidence of the maintenance of its activity. Authors presented antiviral activity data (CPE50) (lanes 435-436) and concluded “accumulation level of native interferon-aA.. is by 40% lower as compared with that for the hybrid protein…”. Two questions:

3a. It is not clear which protein preparation was used for this bioassay: TPL, PLTT or PEP?

  • Added bioassay results for each protein preparation (Figure 5b).

3b. The suggestion that the increase of bioactivity is due to a higher amount of protein may or may not be true, needs some more evidence. The alternative possibility of better protein stability needs to be evaluated.

  • Added interferon bioavailability data, as well as Western blot data before and after purification procedures, and demonstrated a correlation between lichenase activity (this value indicates the amount of recombinant protein in the sample) and interferon bioactivity (the second part of the recombinant protein), as well as an increase in amount of recombinant protein. In our opinion, these data may indicate that the increase in biological activity may be due to an increase in the level of recombinant protein in plant protein lysates.

It should be noted that lichenase has outstanding stability, based on this property, it is used in the food and brewing industry, moreover, based on lichenase, recombinant proteins have been created, with direct fusion and due to the insertion of domains, and the preservation of this property has been demonstrated in a number of studies, some of which we cite in the manuscript.

  1. Comment: If expressed as a fusion protein, a strategy for purification of the protein is necessary for pharmaceutical use, otherwise there is the need for justification for using the fusion protein directly for treatment.
  • Note that any of the used tags for purification can be included in the design that is optimal in terms of efficiency and used for protein purification. We have previously demonstrated this when expressing the same recombinant protein in a bacterial system, the reference is in the manuscript. We performed and presented data from an exploratory study aimed at elucidating additional regulatory contexts and approaches to increase the yield of target proteins in plant systems.

  1. Comment: Figure 4 intends to showcase value of heat treatment or ethanol precipitation for purification of protein but needs some improvements (lanes 481-486):

Its two panels give total soluble protein (panel 1 on the left) and lichenase activity (panel 2 on the right) for A: total protein lysate (TPL); B: protein lysate after temperature treatment (PLTT) after 65oC incubation, proteins after ethanol precipitation (PEP). Blue columns are for control N. benethamiana plants and yellow columns are for the pVIG-T-L53-In-M transformed plants (in reference to its M&M part, lanes 575-586). It is noted that there is no volume data on dissolving ethanol precipitated protein.

  • Done. Section Materials and Methods added information about volume data on dissolving ethanol precipitated protein.

5a. panel 1 suggests that transformed plants have higher total protein concentration, the increase of protein mostly remain soluble after heat treatment. Since the data are represented by concentration (ug/ml), the buffer volume to redissolve ethanol precipitate should be given for better interpretation. A better presentation is the use of protein quantity per fresh weight of leave (ug / mg fresh weight).

  • Done. Section Materials and Methods added information about volume data on dissolving ethanol precipitated protein.

5b. Panel 2, on the other hand, seems to suggest that specific activity for lichenase increased after heat treatment; such increase is even more so (>10x) after ethanol precipitation. Authors suggest (lanes 473-476) that “a considerable decrease in the share of the other total soluble proteins”. This is not convincing enough. Authors need to provide some protein purity data, preferably SDS-PAGE gel staining to show increased protein purity. It is also suggested to show changes in total protein quantity..

  • Done. The electropherogram of proteins in the SM is shown (Figure 4S). It was the total soluble protein that was evaluated based on the following: (1) lichenase, as well as recombinant proteins constructed on its basis, are soluble protein; (2) when isolating a recombinant protein, it was precisely the fractions of soluble proteins that were selected that would include recombinant lichenase with interferon integration (see M&M), removing other proteins in the form of cellular debris. In our opinion, changes in the total amount of proteins during purification by temperature and precipitation with ethanol will not be so significant, although, of course, this requires additional research in the future

We tried our best to improve the manuscript and made some changes in the manuscript, and these changes will not influence the content and framework of the paper.

We appreciate for Reviewer’ warm work earnestly, and hope that the correction will meet with approval. 

Once again, thank you very much for your comments and suggestions.

Best regards.

Yours sincerely,

Round 2

Reviewer 1 Report

The manuscript improved substantially. There is just one comment:

" Note that the inserted domain (IFNαA-M) in L53-IN-M is also 531
able to refold in vitro after heating and ethanol denaturation as evidenced by the results 532 of the antiviral activity of protein plant lysates before and after heating and precipitation 533 with ethanol (Figure 6)." --- I don't think the Western blot in Figure 6 can tell you in vitro refolding of the protein. There is no ladder on the Western blot image.

Author Response

Thank you for your letter and for editor’s and reviewer’s comments concerning our manuscript. Those comments are all valuable and very helpful for revising and improving our paper, as well as the important guiding significance to our research. We have studied comments carefully and have modified the manuscript.

The main corrections in the paper and the responds to the Editor’s and Reviewer’s comments are as flowing:

Editor

First, careful reading is necessary in order to improve English style, the second sentence from the Introduction is incomprehensible. Pay attention to the scientific names of the organisms and expressions in Latin, they must be written in italics: e.g.: Nicotiana benthamiana in line 83, E. coli in line 137, among others.

  • done.

Reviewer#

RESPONSES TO COMMENTS

  1. Comment: The quality of the newly added figure should be improved for better resolution
  • done.

Reviewer##

RESPONSES TO COMMENTS

  1. Comment: " Note that the inserted domain (IFNαA-M) in L53-IN-M is also able to refold in vitro after heating and ethanol denaturation as evidenced by the results of the antiviral activity of protein plant lysates before and after heating and precipitation with ethanol (Figure 6)." --- I don't think the Western blot in Figure 6 can tell you in vitro refolding of the protein. There is no ladder on the Western blot image.
  • done. There was a technical error - instead of figure 5, figure 6 was indicated. Corrected. In Figure 6 (Western blot), a molecular weight marker is added.

We tried our best to improve the manuscript and made some changes in the manuscript, and these changes will not influence the content and framework of the paper.

We appreciate for Editors and Reviewer’ warm work earnestly, and hope that the correction will meet with approval. 

Once again, thank you very much for your comments and suggestions.

Best regards.

Yours sincerely,

Reviewer 2 Report

Clear improvements have been made and most of the issues raised are acceptably addressed. 

The quality of the newly added figure should be improved for better resolution.

Author Response

(The authors gave the same response as above.)
